# EmbedTAD Using Graph Embedding and Unsupervised Learning to Identify TADs from High-Resolution Hi-C Data
H. M. A. Mohit Chowdhury [1,2] & Oluwatosin Oluwadare [1,2] ✉

Topologically Associating Domains (TADs) serve a functional purpose as self-interacting regions whose boundaries are enriched with various proteins. Identifying these TAD regions is essential for examining several biological characteristics, including immune system function and chromosome organization. In this study, we propose EmbedTAD for identifying TAD regions from high-resolution Hi-C data. To achieve this, we utilize NetMF, a graph embedding technique that employs low computational resources, and cluster the embeddings into TAD regions using the HDBSCAN algorithm. We demonstrate that, during T-cell differentiation, EmbedTAD detects TAD rearrangements and can differentiate between active and inactive cells. Furthermore, we show that EmbedTAD recovers a significant number of TADs also present in PLAC-seq data, demonstrating its reproducibility. We confirm that EmbedTAD detects TADs with distinct ChIP-seq signals surrounding their boundaries, including CTCF, RAD21, and SMC3. Overall, EmbedTAD reliably and efficiently identifies TADs with minimal computational resources, outperforming many state-of-the-art methods.

One of the key parts of living beings are chromosomes, which contain genetic and epigenetic information to control functional traits[1]. The genome's three-dimensional (3D) structure is essential for gene regulation, including chromatin interactions that influence phenotypes and contribute to understanding gene expression, as well as the organization of active and inactive territories[2–5]. 3D chromosome conformation capture methods, such as 5C, Hi-C, and Micro-C, enable the mapping of spatial interactions between genomic regions that are distant in linear sequence, providing insights into chromatin structure, gene regulation, and cellular function[6]. These techniques have enabled the study of the 3D spatial structure of chromosomes and genomes[3]. Specifically, Hi-C has revealed that chromosomes are segregated into kilobase to megabase-sized regions, creating physical domains known as Topologically Associating Domains (TADs)[7,8]. According to Dixon et al.[8], there are self-interacting regions with a bin size of less than 100 Kb that form a triangle-like shape and are bounded by segments before the interaction abruptly terminates. These areas are known as TADs, and the sudden shift denotes the boundary areas that divide TADs[8]. TADs are enriched in genes that interact with regulatory elements, and their boundaries are enriched with various epigenetic proteins, including insulator proteins and others[2,8,9]. TADs play a crucial role in defining interacting domains and genomic functions, such as the formation of chromatin loops[10]. They are also important for higher-order chromatin folding and proper long-range transcriptional control[6]. Within a TAD, loci share

common cis-regulatory elements and form interaction networks. During cell development, gene expression patterns are influenced by the physical clustering of TADs, which form a modular framework in chromatin structure and nuclear positioning[7]. In mammalian genomes, TAD regions align with histone protein markers such as H3K27me3, H3K9me2, as well as CTCF and cohesin-binding sites[7,8].

TAD detection has been a prominent field of study, and several computational methods have been developed to identify TADs from Hi-C data, including[11–16]. Researchers have conducted comparative studies and classified these tools into several categories, including linear score based, statistical model based, network feature based, graph partitioning, and clustering based approaches[17–20], in order to benchmark these methods and highlight the importance of TAD detection research. Within these categories, linear score based methods assign a score to each genomic bin's contact frequency and apply statistical testing to identify TADs. This category includes most TAD detection tools, such as Armatus[21] and TopDom[15]. For instance, TopDom computes bin signals using a sliding window and identifies statistically significant domains. The statistical modeling category divides data and variables based on their interaction distributions or relationships, applying statistical tests to filter false-positive TADs. HiCSeg[14], for example, uses dynamic programming and log-likelihood computations to detect TADs. This approach can also be described as feature based modeling, since it leverages interaction features to infer domain boundaries.

[1]Department of Computer Science and Engineering, University of North Texas, Denton, TX, USA. [2]Center for Computational Life Sciences, University of North Texas, Denton, TX, USA. ✉e-mail: oluwatosin.oluwadare@unt.edu

Tools in the graph modeling category transform the Hi-C contact matrix into a graph data structure, where edges represent interaction frequencies between loci. Spectral[16] identifies TADs using spectral graph theory and the Fiedler vector, while scKTLD[22] detects TADs from single-cell Hi-C data through graph embedding and dynamic programming. We refer to graph modeling and network modeling interchangeably, given their conceptual similarity. Lastly, there are clustering based approaches[17–20], which group genomic regions based on interaction similarity to infer hierarchical or non-overlapping domain structures. A number of clustering algorithms have been proposed for TAD detection tools, such as IC-Finder[13], which uses hierarchical clustering to identify TAD domains, ClusterTAD[12], which uses K-means clustering[23], and CASPIAN[11], which uses HDBSCAN[24].

Despite advancements in TAD detection, the use of spatial feature representation of bulk Hi-C data through graph embedding remains limited. HiC-GNN[25] introduced the use of a graph embedding-based approach for 3D genome structure reconstruction from bulk Hi-C data, and HiCEGNN[26] recently proposed a similar approach for single-cell Hi-C data 3D chromosome structure reconstruction. However, the application of graph embedding techniques to TAD identification using bulk Hi-C data has not yet been explored, representing a significant opportunity for methodological innovation. Graph-based representation of Hi-C data offers a powerful means of understanding the complex interactions within TADs. By structuring the data as a graph, we can better capture the relationships and connectivity among genomic features, providing a more nuanced view of spatial organization. Graph embedding techniques retain critical attributes of the data while simplifying computation, making it easier to analyze large datasets.

In this study, we present EmbedTAD, an algorithm that integrates graph embedding and clustering to identify TADs from high-resolution Hi-C contact matrices. EmbedTAD employs an efficient embedding strategy that maximizes feature representation in a low-dimensional space while minimizing computational cost. The accuracy and biological relevance of the TADs detected by EmbedTAD were validated using multiple evaluation metrics and biological datasets. We first compared EmbedTAD's performance with existing TAD callers using both in-silico and in-situ Hi-C datasets. We then conducted a systematic evaluation of the TAD regions identified by EmbedTAD and demonstrated its practical applications in understanding cellular functional organization. The identified TADs facilitate the exploration of key biological features, including CTCF binding sites, T-cell activation during development, and histone modification markers such as H3K27ac and H3K27me3. Overall, EmbedTAD represents Hi-C interaction data through network embedding in a low-dimensional space, achieving low memory usage, reduced runtime, and superior performance compared to other state-of-the-art TAD callers. These characteristics make EmbedTAD an efficient and powerful tool for TAD detection and genomic structure analysis.

## Results

### EmbedTAD overview

To predict TADs, EmbedTAD starts with an $n \times n$ Hi-C contact matrix as input (Fig. 1A). We observed that the size of a single chromosome of a high-resolution Hi-C contact matrix is often too large, particularly for human and mouse data, and most computing systems are unable to process the entire matrix at once. To address this issue, we divided the $n \times n$ contact matrix into $p \times p$ equal-sized sub-matrices following Equation (2). EmbedTAD determines $ns$ (number of sub-matrices) dynamically at runtime according to the size of $n \times n$ and a threshold $t = 5000$, resulting in sub-matrices of size $\leq 5000 \times 5000$. While dividing the $n \times n$ Hi-C contact matrix into equal-sized $p \times p$ sub-matrices, we observed missed boundary cases, degrading performance by 2–3%. To account for this loss, we extended every $p_{i+1} \times p_{i+1}$ sub-matrix by a $q = 3Mb$ region from the previous $p_i \times p_i$ sub-matrix (Fig. 1B). In general, we represent each sub-matrix as $p \times p$ and apply a Gaussian filter to remove noise and strengthen the diagonal interaction frequencies.

Next, we convert each $p \times p$ sub-matrix into graph data to feed into the NetMF embedding algorithm[27]. We used an embedding size of $e = 455$, as it achieved optimal results during our hyperparameter search (see Hyperparameter Search in supplementary information). The embedding reduces the feature size to $e$ while preserving the information present in $p$ features. The embedded data are then fed into the HDBSCAN[24] clustering algorithm to produce clusters, which represent TADs. Because we extended $p_{i+1} \times p_{i+1}$ by $q$, some overlapping or redundant TADs appear in the extended ($Q$) region, which originates from the previous $p_i \times p_i$ sub-matrix (Fig. 1B). To remove these overlapping or redundant TADs, we measure the TAD Quality ($TQ_i$) for $p_i \times p_i$ and $TQ_{i+1}$ for $p_{i+1} \times p_{i+1}$ in the $Q$ region. We retain the set with the highest TAD quality.

Iteratively, EmbedTAD applies a Gaussian filter to each sub-matrix ($p \times p$), creates a graph, feeds it into the NetMF and HDBSCAN algorithms, and removes overlapping or redundant TADs to produce TADs for the entire matrix.

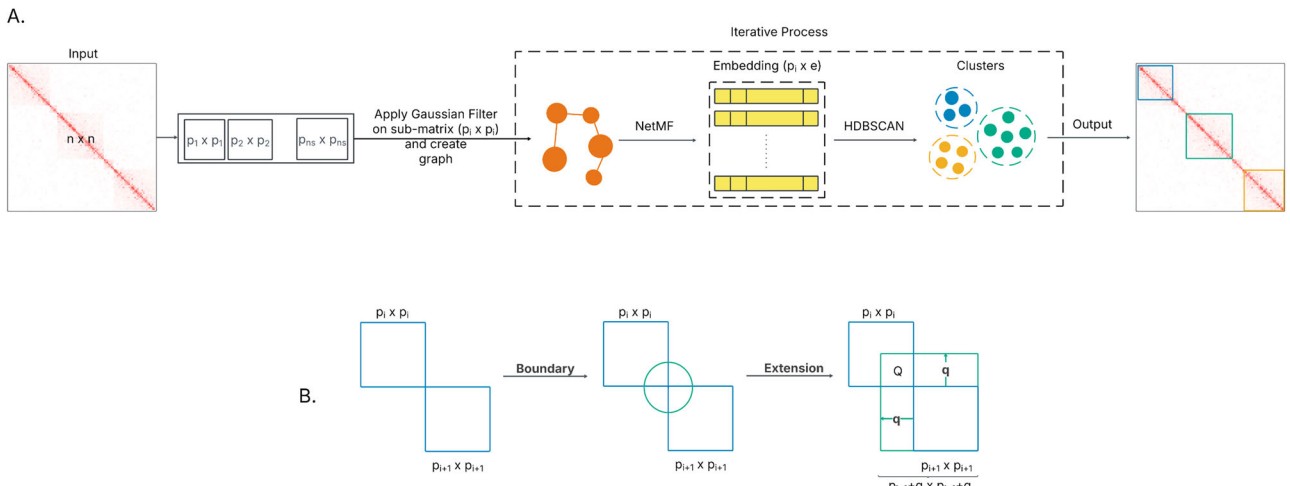

**Fig. 1 | Overview of EmbedTAD. A** A $n \times n$ Hi-C matrix is processed through the EmbedTAD pipeline, which (i.) splits it into equal-sized $p \times p$ sub-matrices, (ii.) converts each sub-matrix into a graph representation, (iii) embeds the graph using the NetMF algorithm, and (iv) clusters the embedded data with the HDBSCAN algorithm to identify TADs. Steps (ii) through (iv) are repeated for each submatrix, and the resulting clusters collectively yield the TAD regions identified in the original $n \times n$ Hi-C matrix. **B** Extension of the sub-matrix, $p \times p$ by $q = 3Mb$ region resulting $(p_{i+1} + q) \times (p_{i+1} + q)$ sub-matrix to cover the boundary cases. By using Equation (2) to divide the $n \times n$ Hi-C matrix into an equal-sized $p \times p$ sub-matrix, we found that we could lose possible interaction in the boundary regions (bluish green circle). We enlarged the $p_{i+1} \times p_{i+1}$ region by $q = 3Mb$ within the $p_i \times p_i$ sub-matrix in order to encompass this boundary region (top and left bluish green rectangles).

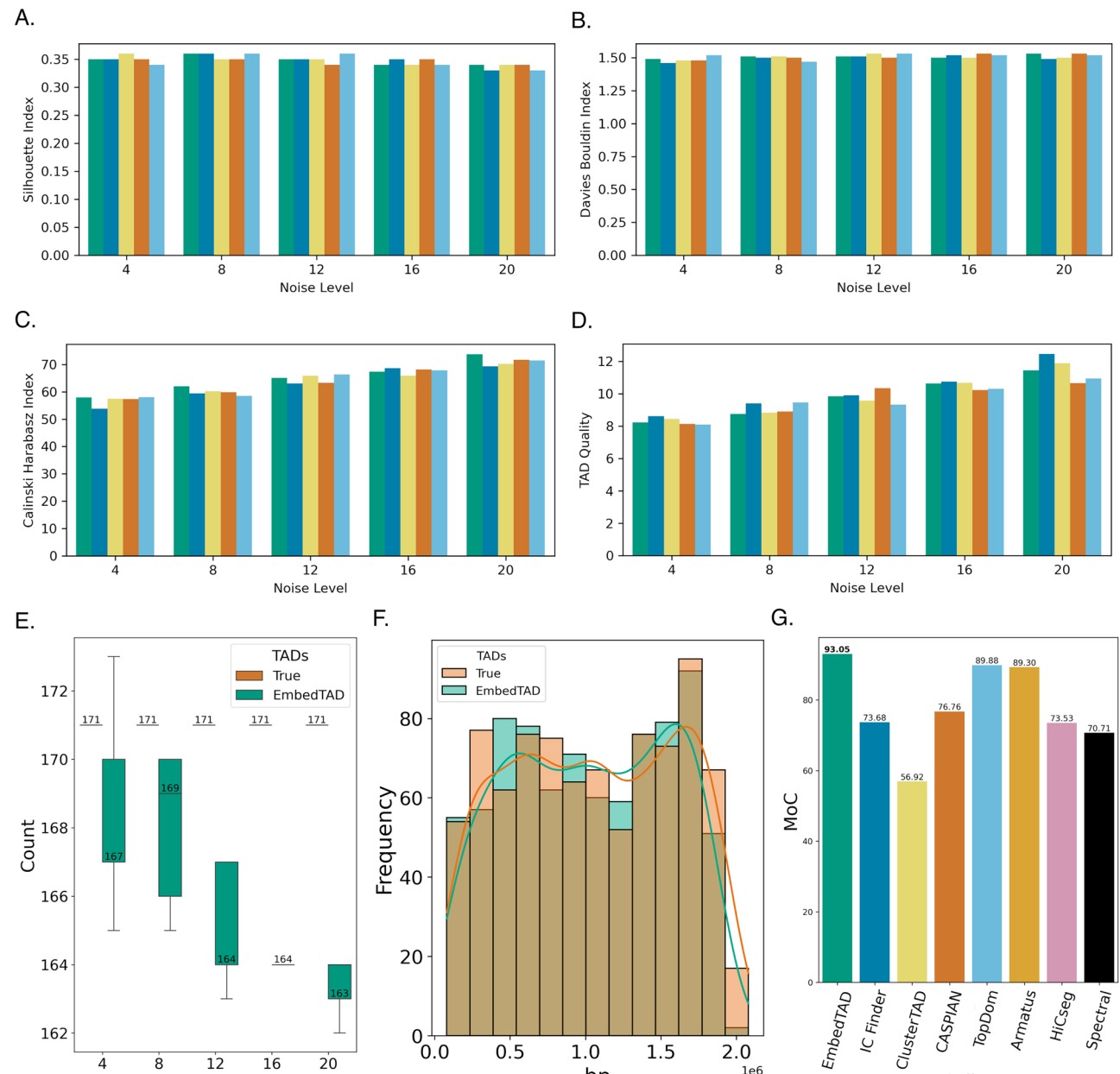

**Fig. 2 | EmbedTAD's consistency analysis using In-silico Hi-C data.** EmbedTAD showed consistent. **A** Silhouette Index (SI), **B** Davies-Bouldin Index (DBI), **C** Calinski-Harabasz Index (CHI), and **D** TAD Quality scores across all noise levels (4, 8, 12, 16, and 20) using In-silico Hi-C data (at a given noise level in the x-axis, a distinct color denotes different data). EmbedTAD scores for every metric are nearly close for all data for a particular noise level (e.g., all datasets in 8 noise level). EmbedTAD showed nearly similar scores in SI and DBI for all data across all noise levels, a gradual improved score in CHI and TAD quality for increasing noise level. **E** EmbedTAD showed consistent results in True vs. detected TAD count across all noise levels using In-silico Hi-C data. It detected nearly similar number of TADs (median) across all noise levels. **F** EmbedTAD showed consistent results in True vs. detected TAD size distribution across all noise levels using In-silico Hi-C data. The ground-truth curve and its kernel density estimate (KDE) curve are close to one another, with the majority of TADs ranging between 250Kb and 1.75Mb in size. **G** EmbedTAD achieved the highest Measure of Concordance (MoC) score compared with other TAD callers at different noise levels using 455 embedding size.

## EmbedTAD shows consistency across different noise level using In-silico Hi-C data

We observed the consistency of EmbedTAD across different noise levels (4, 8, 12, 16, and 20) using the In-silico Hi-C dataset[28]. We computed the Silhouette Index (SI), Davies-Bouldin Index (DBI), and Calinski-Harabasz Index (CHI) to evaluate cluster quality and consistency across different noise levels (Fig. 2A, B, and C). EmbedTAD produced a consistent SI ($\approx$ 0.35) and DBI ($\approx$ 1.50) across different noise levels. The CHI score fluctuated between 55 and 70 across these noise levels. In addition, we computed the TAD Quality (TQ)[12] for each dataset and found that the scores were consistent among the five datasets at each noise level. All noise levels maintained TQ scores between 8 and 12 (Fig. 2D). We also observed the number of detected TADs at each noise level and compared the results with the ground-truth. EmbedTAD's detected TAD count was slightly lower than the true TADs, with the median value closely matching the true count of 171 TADs across all noise levels (Fig. 2E). To further validate EmbedTAD's consistency across different noise levels using the In-silico Hi-C dataset, we measured the TAD size distribution. We observed that EmbedTAD's kernel density estimation (KDE) curve was almost consistent with the ground-truth curve, with most TAD sizes falling between 250 Kb and 1.75 Mb (Fig. 2F and Supplementary Fig. 1). Overall, EmbedTAD showed consistent results across all metrics using the In-silico Hi-C dataset.

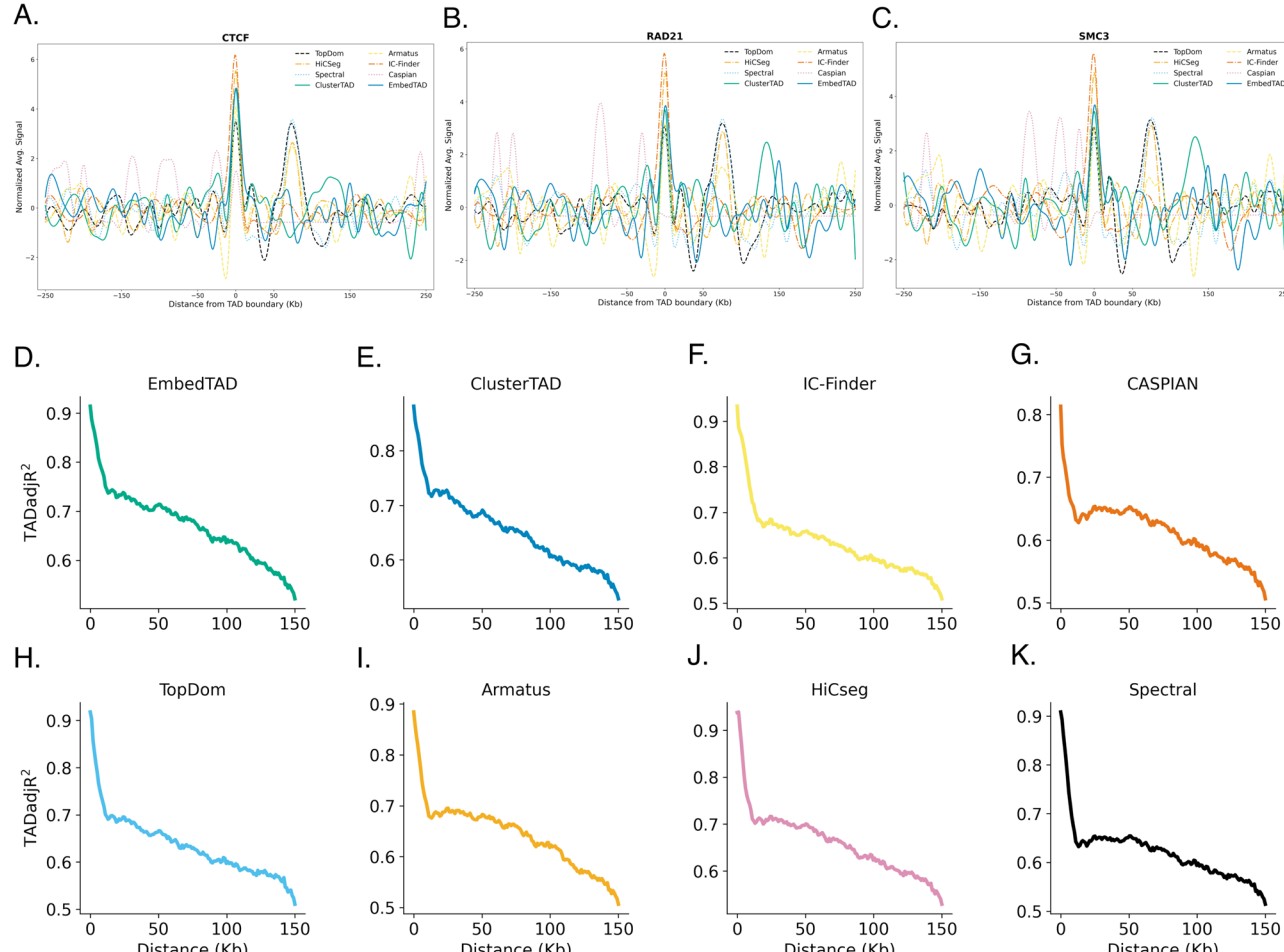

**Fig. 3 | Comparison of TAD callers using GM12878 chromosome 19 at 10Kb resolution.** Comparison of **A** CTCF, **B** RAD21, and **C** SMC3 average ChIP-seq signal in ± 250$Kb$ region with other TAD callers where EmbedTAD showed enrichment signal at boundary. TAD boundaries are enriched with CTCF and cohesin proteins, and depletion of these proteins weaken TAD insulation leading to a weakened boundary. **D**–**K** Line plots explains the interaction frequency variance over the region (0 to 1.5Mb) using $TADadjR^2$ metric on GM12878 chromosome 19 at 10Kb resolution across other state-of-the-art TAD callers. EmbedTAD obtained a score that is near to 1 and went down with increasing genomic distance, like with other state-of-the-art TAD callers. This high score and depletion over distance offer the ideal explanation for variation in interaction frequency and are crucial for distinguishing between TAD and non-TAD regions.

## EmbedTAD outperforms across all noise level using In-silico Hi-C data with other TAD callers

We performed a thorough comparison of EmbedTAD with seven state-of-the-art methods to assess how well our pipeline performs relative to other state-of-the-art algorithms. Comprehensive reviews by[17–20] categorized TAD detection algorithms into different groups. To ensure a fair comparison, we randomly selected at least one tool from each category. Since our algorithm falls under the clustering-based category, we included three clustering-based algorithms in our comparison. Specifically, we selected IC-Finder[13], ClusterTAD[12], TopDom[15], Armatus[21], HiCseg[14], CASPIAN[11], and Spectral[16], categorizing them into Clustering, Linear, Statistical, and Network Feature methods based on their underlying approaches (Supplementary Table 1). We extensively compared EmbedTAD with these seven state-of-the-art TAD callers using in-silico Hi-C data across five noise levels (4, 8, 12, 16, and 20) and recorded the Measure of Concordance (MoC)[29] to evaluate performance (Fig. 2G and Supplementary Fig. 2). EmbedTAD consistently achieved the highest MoC scores across all noise levels compared to other TAD callers. We further evaluated its performance using traditional clustering metrics, including Adjusted Mutual Information (AMI) and Adjusted Rand Index (ARI) (see Performance Comparison using AMI and ARI in Supplementary Information), following prior studies in TAD detection[22,30]. EmbedTAD achieved competitive AMI and ARI scores, showing stable and reproducible performance with nearly identical

mean and median values and no outliers (Supplementary Fig. 3, Supplementary Table 2). The results indicate that the network embedding and graph-based data representation reduce noise sensitivity and enhance overall stability. Collectively, these findings demonstrate that EmbedTAD reliably and accurately identifies TADs across varying noise conditions, establishing it as a robust and effective TAD caller.

## EmbedTAD demonstrates robust ChIP-seq signal enrichment at TAD boundaries using In-situ Hi-C data

We evaluated TAD boundaries on GM12878 chromosome 19 and CH12LX chromosome 18 at 5 Kb and 10 Kb resolutions and compared EmbedTAD with other TAD callers. CCCTC-binding protein (CTCF), RAD21, and SMC3 are known to be abundant at TAD boundaries[8,31]. Cohesin proteins, such as RAD21 and SMC3, aid in DNA loop formation and chromosomal structure maintenance, while CTCF functions as an insulator protein. These proteins are often used as markers to evaluate a TAD caller's accuracy and to validate the detected TAD boundaries. We computed the average ChIP-seq signal for all TAD callers listed in Supplementary Table 1 and compared the results with those of EmbedTAD. Specifically, we measured the average ChIP-seq signal at the boundary positions and their ± 250 Kb neighboring regions. In Fig. 3A, EmbedTAD showed CTCF enrichment around boundaries on GM12878 chromosome 19 at 10 Kb resolution. Similarly, when evaluating RAD21 (Fig. 3B) and

SMC3 (Fig. 3C), EmbedTAD showed enrichment around TAD boundaries similar to other TAD callers.

When comparing CH12LX chromosome 18 at both 5 Kb and 10 Kb resolutions, as well as GM12878 chromosome 19 at 5 Kb resolution, we observed a consistent pattern where EmbedTAD showed enrichment signals at boundary positions (Supplementary Figs. 4, 5, and 6). EmbedTAD consistently showed enrichment around boundaries, similar to other TAD callers across analyses. We also note that methods detecting a larger number of TADs (Supplementary Fig. 7) naturally have a higher likelihood of capturing additional boundary variations, which partly explains why the other methods showed slightly higher enrichment of CTCF, RAD21, and SMC3. Taken together, these results underscore the robustness of our algorithm and provide a clear explanation of ChIP-seq signal enrichment around TAD boundaries, which is essential for accurately identifying TADs and maintaining biological information for future analysis.

### EmbedTAD demonstrates impressive TAD detection accuracy implied by *TADadjR²*

TADs are characterized by higher interaction frequencies within the region compared to regions outside, making them significant in cell biology[8]. In contrast, regions with low interaction frequencies are typically identified as non-TAD regions. Interaction frequency generally decays in proportion to distance; within a TAD, local interaction frequencies should remain high and gradually decay as the distance increases. $TADadjR^2$ score (Equation (1)) is an $R^2$-based statistical metric, adjusted for TAD analysis, that explains the variance of Hi-C interaction frequencies over a given genomic distance.[32]. An et al.[32] described $TADadjR^2$ as

$$R_{adj}^2 = 1 - \frac{\frac{1}{N-N_t-1}\sum_{i=1}^{N}\left(X_i - \hat{X}_i\right)^2}{\frac{1}{N-1}\sum_{i=1}^{N}\left(X_i - \overline{X}\right)^2} \qquad (1)$$

where, $X_i = i^{th}$ bin pair contact frequency, $N$ = number of bins pairs, $N_t$ = number of TADs whose size is greater than or equal to this genomic distance, $\hat{X}_i$ = average contact frequency within this TAD or gap region, and $\overline{X}$ = mean contact frequency. This metric explains the interaction frequency variance at a given genomic distance. The numerator calculates the variance of interaction frequency of TADs, and the denominator calculates the overall variance, and this metric ensures that adding more TADs that are not substantially significant will not affect the score by penalizing the TADs' variance with overall variance. As TADs have a high local interaction frequency, and this frequency decays over the distance at the boundary region, this metric will quantify the detected TADs with interaction frequency explanation. It generates scores from 0 to 1; close to 1 means the classified TADs perfectly explains the interaction frequency variance and decays proportionally with increasing distance, and 0 means no explanation of frequency variance.

To assess TAD detection accuracy, we measured the $TADadjR^2$ score from 0 to 1.5 Mb region for each TAD caller and compared it with EmbedTAD. We plotted the $TADadjR^2$ scores in Fig. 3D to K and Supplementary Figs. 8, 9, and 10 to visualize the decay pattern over the region. EmbedTAD showed high interaction frequency, and this interaction frequency depleted over increasing distance, like other state-of-the-art TAD callers. Along with other state-of-the-art TAD callers, EmbedTAD obtained a score that is close to 1 and decreased over increasing genomic distance, as indicated in Supplementary Table 3. This high score and deplation over distance are essential for differentiating between TAD and non-TAD regions and provides a perfect explanation for interaction frequency variation. EmbedTAD's maintains a small difference in numerical score with other state-of-the-art TAD callers which indicates a strong agreement with others. This analysis of Hi-C interaction frequency variance demonstrates EmbedTAD's capability to distinguish between TAD and non-TAD regions.

### EmbedTAD accurately detects TADs with strong interaction signals, consistent sizes, and insulation scores

We statistically analyzed the TADs detected by EmbedTAD on GM12878 and CH12LX at 5 Kb and 10 Kb resolutions. In Fig. 4A, we plotted the total number of TADs detected in each dataset and observed that EmbedTAD detected more TADs at 10 Kb resolution compared to 5 Kb. We also examined the TAD size distribution per bin and found that the average size ranged between 200 Kb and 280 Kb, indicating that the typical TAD size detected by EmbedTAD lies within this range (Fig. 4B, C, D, E, and F). To further evaluate the detected TADs, we visualized[33,34] GM12878 chromosome 19 (Fig. 4G) and CH12LX chromosome 18 (Fig. 4H) at 10 Kb resolution, focusing on the 40 Mb to 44 Mb region. The TADs are shown with blue lines at the top, and the insulation score below these TAD regions indicates the Hi-C interaction frequency strength. The TAD cutoff (red line) signifies regions with strong Hi-C interaction frequency, a fundamental feature of TAD boundaries[34]. We further examined the insulation scores on GM12878 chromosome 3 (5 Kb and 10 Kb), chromosome 19 (5 Kb), and CH12LX chromosome 2 (5 Kb and 10 Kb) and chromosome 18 (5 Kb), and observed that the detected TAD regions had insulation scores above the TAD cutoff in most cases (Supplementary Fig. 11 and 12). Overall, EmbedTAD detected the majority of TADs with strong Hi-C interaction frequencies and insulation scores exceeding the TAD cutoff.

### EmbedTAD achieves efficient TAD detection with low memory and fast execution

Running time and memory consumption are crucial factors for any algorithm, especially when processing high-volume genomic data. We performed a detailed performance comparison of CPU and GPU implementations of EmbedTAD (see Performance Comparison of CPU and GPU Implementations of EmbedTAD in supplementary information). While both versions are available we adopted the GPU implementation of EmbedTAD as the default. We recorded the running time (in seconds) and memory consumption (in MB) for each TAD caller on GM12878 chromosome 19 and CH12LX chromosome 18 at 5 Kb and 10 Kb resolutions to evaluate the performance of EmbedTAD compared to other TAD callers. As shown in Supplementary Table 4, EmbedTAD ranked as the third fastest algorithm in terms of execution time. In terms of memory consumption (Supplementary Table 5), EmbedTAD used the least memory among all the evaluated methods. While running time and memory usage are important for usability, EmbedTAD also maintains high accuracy in detecting TAD regions, offering an efficient and accurate solution.

### CTCF binding and histone modifications validate TAD detection in EmbedTAD

CTCF binding protein is known to be enriched at TAD boundaries. Dixon et al.[8] investigated factors contributing to TAD formation and discovered that 15% of TAD boundaries contain CTCF binding sites. TAD and its' boundary regions should also be assessed for other elements, such as histone modifications and transcription factors[8]. To confirm the biological relevance of the TADs detected by EmbedTAD, we analyzed TAD regions using markers such as H3K27ac, H3K27me3, H3K4me1, H3K4me3, and H3K9me3. Visualization[35] of Hi-C interactions alongside ChIP-seq signals validated that the detected TADs are enriched with these proteins. We observed that active enhancers, such as H3K27ac, and gene body markers, like H3K4me3, are enriched within TAD regions in GM12878 chromosome 19 (Fig. 5A) and CH12LX chromosome 18 (Fig. 5B) at 10 Kb resolution, while repressive marks such as H3K9me3 are not enriched around TADs.

To further validate EmbedTAD's results, we performed this analysis on GM12878 chromosomes 3 and 19, and CH12LX chromosomes 2 and 18 at both 5 Kb and 10 Kb resolutions (Supplementary Figs. 13-18). We consistently observed that TAD regions detected by EmbedTAD are enriched with active enhancers and depleted of repressive markers. Overall, this analysis demonstrates EmbedTAD's ability to detect TADs accurately while preserving key biological features.

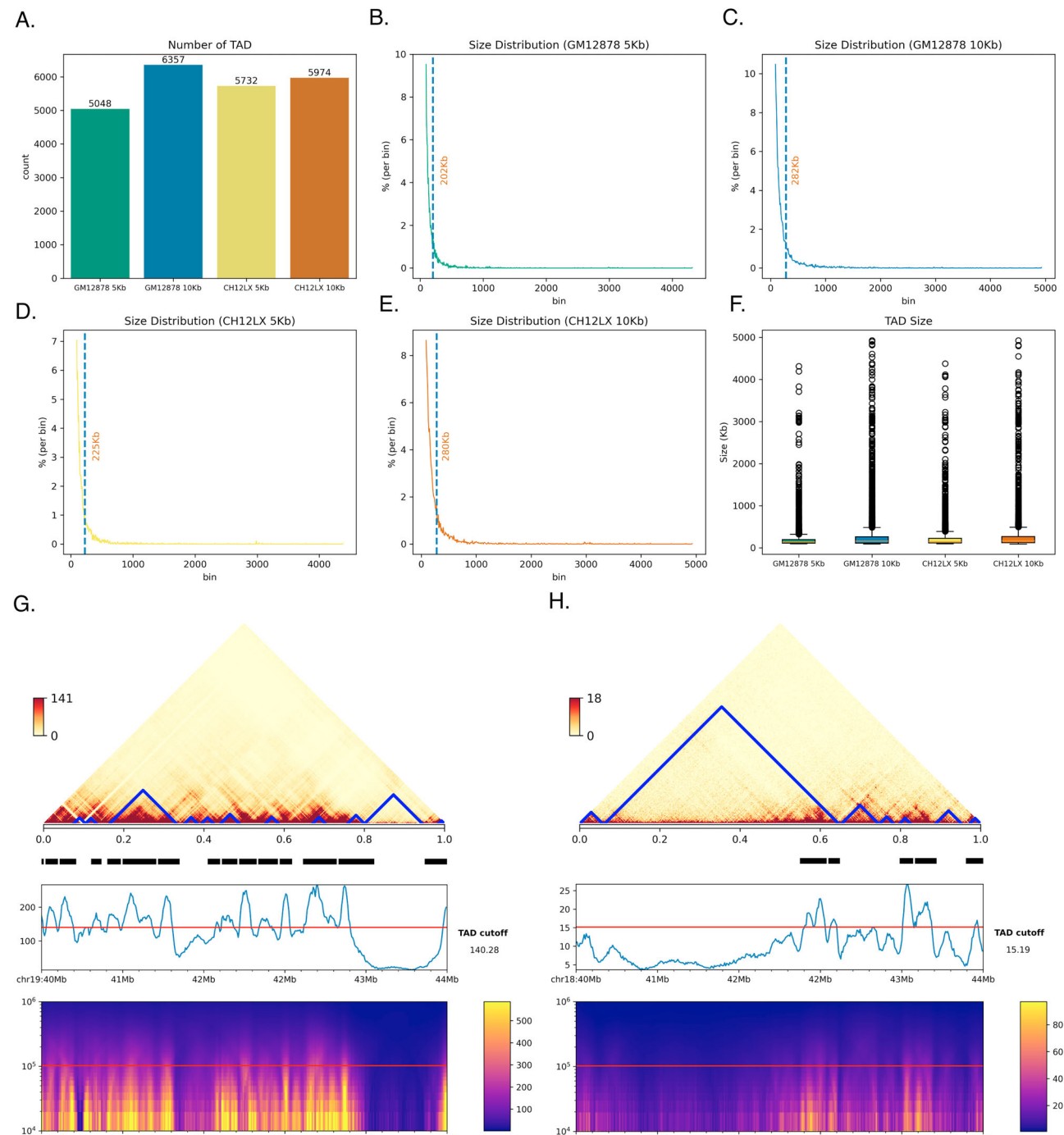

**Fig. 4 | Statistical analysis of EmbedTAD using in-situ Hi-C data. A** EmbedTAD detects more TADs at 10Kb resolution compared to 5Kb resolution from same organism (GM12878 and CH12LX). **B–E** TAD Size Distribution at 5Kb and 10Kb resolution using GM12878 and CH12LX. Average TAD size (blue line) of both organisms is nearly same at the same resolution, whereas EmbedTAD discovered larger TAD at 10Kb resolution than at 5Kb resolution. **F** Box-plot shows TAD size using GM12878 and CH12LX at 5Kb and 10Kb resolution where EmbedTAD showed consistent results between 5Kb and 10Kb resolution. This visualization also validate previous distribution plot prediction. Insulation score (IS) using **G** GM12878 chromosome 19 and **H** CH12LX chromosome 18 at 10Kb resolution between 40Mb and 44Mb regions. EmbedTAD's detected TADs are annotated with blue lines (top), line plot shows IS over the region with TAD cutoff threshold and bars indicate currently called TADs (middle), and heatmap with IS score (bottom). Within this region, EmbedTAD showed significant agreement between detected TADs and IS scores.

## EmbedTAD recovers TAD regions from mESC H3K4me3 PLAC-seq data

A major challenge for TAD callers is the lack of accessible ground truth, leading to variations in TAD identification across different methods. While the majority of TAD regions should overlap, even when window size is taken into account, these regions often differ between TAD callers. This overlap can be described as the reproducibility among TAD callers. Another significant challenge is that most TAD callers are based on bulk Hi-C data, where genes are enriched with proteins, including the active enhancer mark H3K4me3.

In this study, we identified TADs using PLAC-seq data, as researchers have demonstrated its application for 3D genome analysis[36,37]. PLAC-seq

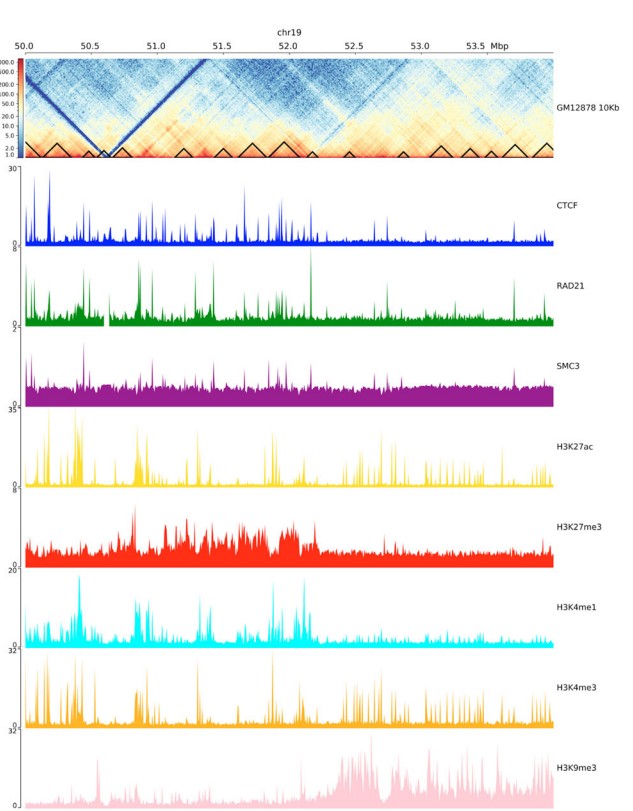

**Fig. 5 | Biological validation of EmbedTAD using different ChIP-seq signal data.** Visualization of EmbedTAD identified TADs on **A** GM12878 chromosome 19 and **B** CH12LX chromosome 18 at 10Kb resolution with different ChIP-seq signals (CTCF, RAD21, SMC3, H3K27ac, H3K27me3, H3K4me1, H3K4me3, and H3K9me3) from 50Mb to 54Mb region. The observed TADs' enrichment with these proteins has been supported by the visualization of Hi-C interactions in conjunction with ChIP-seq results. We observed that repressive marks like H3K9me3 are not enriched around TADs, while active enhancers like H3K27ac and gene body markers like H3K4me3 are.

---

generates a higher proportion of long-range intra-chromosomal pairs (67%) and fewer inter-chromosomal pairs (11%)[38]. In their study, Fang et al. (2016)[38] demonstrated that PLAC-seq improves both efficiency and accuracy over ChIA-PET in detecting long-range chromatin interactions in mammalian cells, generating reproducible contact maps across biological replicates. In mouse embryonic stem (ES) cells, PLAC-seq successfully captured promoter-centered interactions, and H3K4me3 PLAC-seq proved useful for identifying chromatin interactions at active or poised promoters[38]. Together, these findings established PLAC-seq as a reliable method for mapping long-range chromatin interactions. Subsequent studies have further validated the use of PLAC-seq for 3D genome analysis. For example, Lee et al.[37] used H3K27ac PLAC-seq data to demonstrate that TADs and sub-TADs constrain enhancer-promoter interactions, highlighting the utility of PLAC-seq for structural domain analysis. Similarly, Rosen et al.[36] developed the HPTAD method, specialized for TAD detection using PLAC-seq data. They benchmarked TADs detected from PLAC-seq against Hi-C-derived TADs and showed that PLAC-seq can support domain-level analysis.

Since Rosen et al.[36] developed cutting-edge techniques for identifying TADs using mESC H3K4me3 PLAC-seq data, we utilized their TADs as the ground truth to evaluate our method. We calculated EmbedTAD's recovery rate using its detected TADs from mESC bulk Hi-C data. As shown in Fig. 6A, EmbedTAD recovered approximately 67% of the TAD regions identified in the PLAC-seq data, with the exception of chromosomes 15 and 18 at 40 Kb resolution. While our method successfully recovered TADs on most chromosomes, it struggled on chromosomes 15 and 18 where PLAC-seq coverage was particularly sparse, leading to weaker domain signals. Because PLAC-seq targets specific proteins such as H3K4me3, the resulting data are inherently sparser than Hi-C and better suited for detecting chromatin loops (e.g., promoter-enhancer and promoter-promoter interactions) rather than large-scale 3D domains[38]. This reflects a limitation of the input PLAC-seq data rather than the algorithm itself, as bulk Hi-C provides the broader coverage needed for robust TAD detection and chromosome structure analysis. We further compared detected TAD regions from chromosome 1 (Fig. 6B), chromosome 3 (Fig. 6C), chromosome 17 (Fig. 6D), and chromosome 19 (Fig. 6E) between 20 Mb and 26 Mb, along with the H3K4me3 ChIP-seq signals. We found that EmbedTAD's detected TADs generally agreed with the mESC H3K4me3-detected TADs, often showing either multiple smaller TADs or similar TAD regions.

This analysis confirms the agreement between PLAC-seq and bulk Hi-C TADs, demonstrating the reproducibility of EmbedTAD and its ability to retain biologically relevant information consistent with PLAC-seq data.

## EmbedTAD identifies TAD rearrangement during T-cell differentiation in mouse cell

T-cell activation significantly impacts both the immune system response and the dynamics of chromatin structure[39]. These cells play key roles in energy production, cell cycle progression, biosynthesis, and other biochemical processes. Naïve CD4+ T-cells undergo differentiation into various helper T-cell subsets, including Th17, Th1, and others, which are involved in autoimmunity, tissue inflammation, and other cellular processes. Zhang et al.[40] demonstrated that TAD rearrangement is one of the significant organizational changes occurring during T-cell differentiation.

In this study, we examined mus musculus Naïve CD4+ (non-active) and Th17 and Th1 (active) T-cells on chromosome 2 at 10 Kb resolution, focusing on properties such as TAD size distribution and structural changes

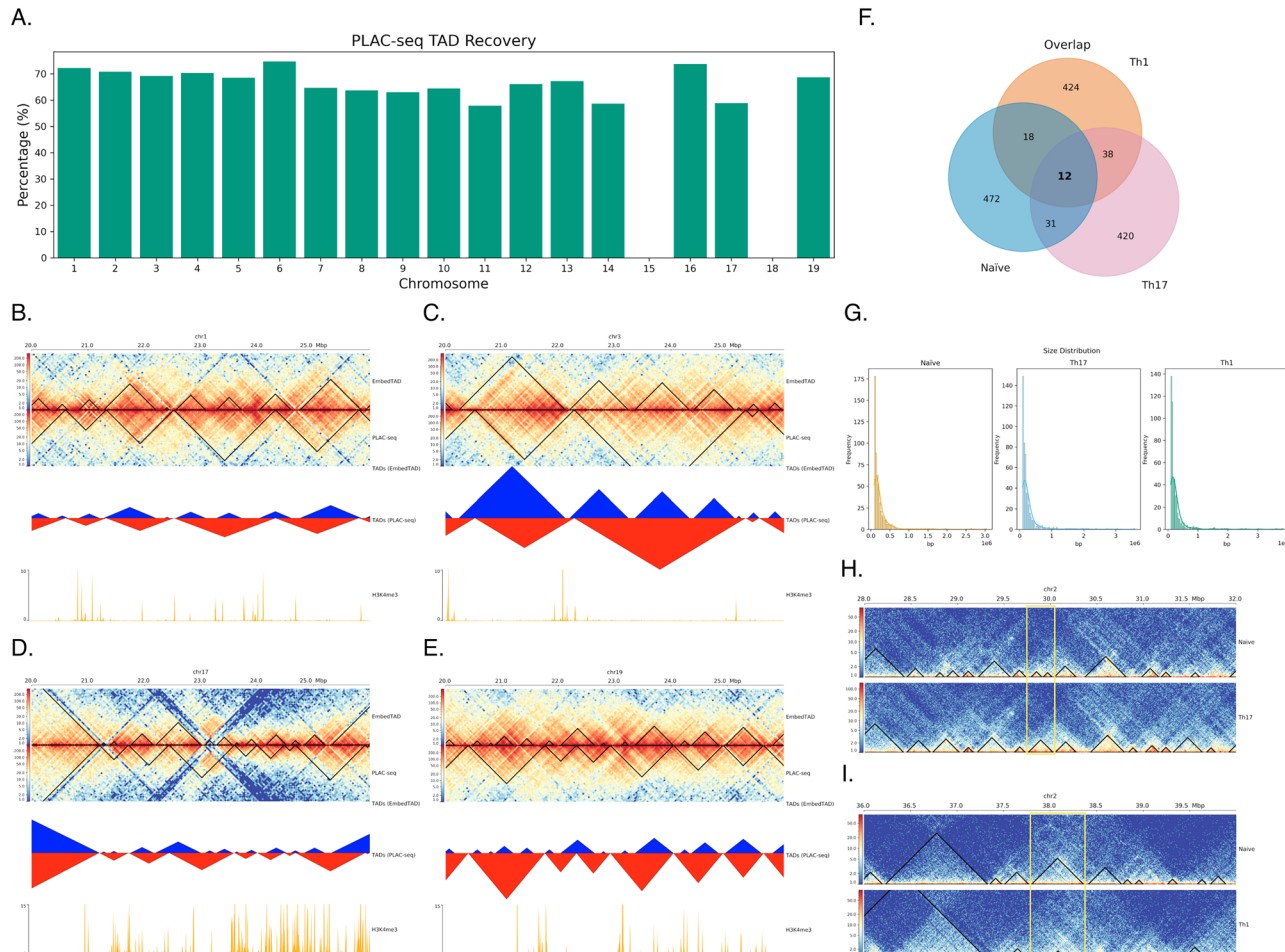

**Fig. 6 | TADs recovery rate using mESC H3K4me3 PLAC-seq data and TAD rearrangement during T cell differentiation. A** EmbedTAD TAD recovery rate across chromosome 1 to 19 at 40Kb resolution using mESC H3K4me3 PLAC-seq data, taking as ground truth the TADs identified by Rosen et al.[36]. Visualization of EmbedTAD identified TADs on **B** chromosome 1, **C** chromosome 3, **D** chromosome 17, and **E** chromosome 19 between 20Mb to 26Mb regions where EmbedTAD overlaps with almost all TADs identified using PLAC-seq data by Rosen et al.[36]. Within these plots **B**, **C**, visualize heatmap with identified TADs with black line (top), shows overlap of TADs detected from both algorithms including EmbedTAD in blue color (middle), and H3K4me3 ChIP-seq signla plot to validate TADs (bottom). **F** EmbedTAD detected TADs overlap using mus musculus on chromosome 2 at 10Kb resolution during T cell (inactive: Naïve, active: Th1 and Th17) differentiation and observed most TADs overlapped between the active cells. **G** Size distribution analysis with EmbedTAD detected TADs from mus musculus during T cell differentiation with kernel density estimate (KDE) curve and observed size changes from inactive to active T-cells. Visualization of **H** Split and **I** Merge event in mus musculus on chromosome 2 at 10Kb resolution during T cell differentiation. In both plot **H**, **I**, the top heatmap with black line indicates TADs during inactive states, the bottom heatmap indicates TADs during active state (Th17 and Th1 respectively), and split and marge event positions are indicated with yellow rectangle.

to validate the differences between active and non-active T-cells. We analyzed overlapping TADs among the three cell types and observed that most TADs overlapped between the active cells (Fig. 6F). We also examined TAD size distribution across the three cell types using KDE curves, revealing size changes from inactive to active T-cells (Fig. 6G). Additionally, we identified and visualized Merge and Split events in TAD domains, finding 21 Split events, and 31 and 28 Merge events in Th17 and Th1 cells, respectively. We highlighted a Merge event in the Th17 cell from the 28 Mb to 32 Mb region (Fig. 6H) and a Split event in the Th1 cell from the 36 Mb to 40 Mb region (Fig. 6I).

Overall, EmbedTAD demonstrates the capability to detect TADs in both active and non-active T-cells, providing insights into their functions and behavior during differentiation.

## Discussion

TADs are important biological features involved in cell development and gene regulation. It is well established that more interactions are observed within TAD regions, and these interactions decay proportionally with increasing distance between loci. TAD boundaries are enriched with different proteins, such as CTCF, which contributes to DNA looping, active enhancer-promoter interactions, and cohesin proteins that help maintain chromosome structure during cell development.

In this study, we developed a pipeline, EmbedTAD, which employs graph representation, embedding, and clustering techniques while using minimal memory and providing fast, robust performance across different organisms and resolutions. EmbedTAD demonstrates competitive performance in various analyses, including MoC scores, ChIP-seq signal enrichment, and reduced memory usage and running time compared to other TAD callers. In addition to achieving strong performance, TADs detected by EmbedTAD were validated through multiple statistical analyses, including agreement in average TAD size distributions and insulation scores. Although EmbedTAD was primarily developed for bulk Hi-C data, it also shows good agreement with TADs detected from PLAC-seq data, further establishing its reproducibility. Moreover, EmbedTAD effectively detects TAD rearrangements during T-cell differentiation, providing valuable insights into the distinction between active and non-active T-cells. By

leveraging graph embedding techniques, EmbedTAD enables the preservation of important genomic features in lower-dimensional space, highlighting its potential for future research in graph-based genomic data analysis. Overall, EmbedTAD utilizes network embedding to represent bulk Hi-C data in a low-dimensional space. To the best of our knowledge, it is the primary method of its kind, achieving competitive running time while consuming the least memory, and demonstrates significant performance compared to other state-of-the-art TAD callers using Hi-C data across different organisms and resolutions, while preserving key biological features for downstream analyses.

## Methods

In this work, we formulate the TAD detection problem as a graph-based problem. A graph data structure preserves relational information between neighboring nodes, where edges define the spatial proximity between two nodes. The distance between nodes indicates their degree of closeness. Additionally, a node, denoted as the central node ($n_c$), is defined with respect to other nodes by establishing a threshold distance value (magnitude), $d_c$, for a specific set of nodes relative to the central node[11,41]. All nodes that satisfy the condition $d < d_c$ (with $n_c$ nodes in total) form a neighborhood, where each node is connected based on spatial proximity. Using this concept, we can derive multiple neighborhoods, with each neighborhood sharing certain properties and collectively forming a cluster. We applied this neighborhood-based graph clustering approach by treating each bin as a node, with the interaction frequency between two bins representing the weight of the edge between them. The Hi-C interaction frequency defines the proximity between bins, where a higher interaction frequency indicates a greater likelihood of interaction. This interaction frequency reveals hidden clusters, with closely related bins forming neighborhoods, which we define as TAD regions. To discover these domains, we developed EmbedTAD (Fig. 1A), which consists of three modules: i) Data Preprocessing, ii) Clustering, and iii) Output. The following subsections provide a detailed description of EmbedTAD.

### Data Preprocessing

As the initial step of EmbedTAD's pipeline, we represented the Hi-C interaction matrix as a graph data structure to feed into the Clustering module. We used in-silico (simulated) Hi-C contact matrices[28] at different noise levels (4, 8, 12, 16, and 20) to determine the hyperparameters (see Hyperparameter Search in supplementary information) for our proposed pipeline and compared EmbedTAD with seven state-of-the-art TAD callers. To evaluate robustness, we also used in-situ (real) Hi-C data[3,31].

Initially, we converted the Hi-C data into an $n \times n$ contact matrix. This contact matrix contains the interaction frequencies between bins ($i, j$). To achieve memory efficiency, we divided the $n \times n$ matrix into smaller sub-matrices of size $p \times p$. EmbedTAD compares the total number of bins with a threshold of 5000 bins. If the total bin count exceeds this threshold, the algorithm determines the optimal number of bins, which is used to divide the entire matrix into smaller sub-matrices. These sub-matrices make the data easier to process without overwhelming memory.

To determine the optimal number of sub-matrices, we used the following equation:

$$ns = \left\lceil \frac{t_{bins}}{t} \right\rceil \tag{2}$$

where $ns$ = number of sub-matrices, $t_{bins}$ = total number of bins, and $t$ = threshold. We then determined the sub-matrix shape as $(p \times p) = (\lceil \frac{t_{bins}}{ns} \rceil \times \lceil \frac{t_{bins}}{ns} \rceil)$.

Since dividing the entire matrix into smaller sub-matrices may disregard boundary regions, we expanded each $p_{i+1} \times p_{i+1}$ sub-matrix on the top and left by $q$ = 3 Mb to accommodate border cases. This $Q$ region was added from the previous $p_i \times p_i$ sub-matrix (Fig. 1B). This resolves the issue of potentially losing TADs at the boundaries. In general, we denote each sub-matrix as $p_i \times p_i$.

After this step, we normalized the contact matrix using a Gaussian filter to remove potential noise. Applying the Gaussian filter improved the performance of our pipeline by 15%–30%. The Gaussian filter normalizes the interaction frequencies, with stronger interactions concentrated along the central diagonal.

Finally, we created the graph structure to feed into the Clustering module to determine the clusters.

### Clustering

The Clustering module is a key component of EmbedTAD's pipeline, where we apply graph embedding and clustering to identify TADs (Fig. 1A). Each cluster is defined as one TAD. In the data preprocessing step, we represent the Hi-C contact matrix as a graph. Proper representation is crucial, as the graph structure contains all the necessary information; without it, processing the data becomes computationally expensive. Embedding addresses this by representing the interaction frequencies of each node as a vector. For instance, a node $i$ may have varying contact frequencies with neighboring nodes $x$ (left) and $y$ (right), and these nodes may have different connections with other nodes, each with distinct contact frequencies. To address this complexity, we applied the NetMF algorithm[27] with varying embedding size and an embedding size of $e$ = 455 produced optimal result (see Determining Optimal Embedding Size and Validation of Optimal Embedding Size Using TAD Quality Metric in supplementary information). NetMF uses network matrix factorization to embed the graph into a lower-dimensional space while capturing the most important features.

Once the data embedding is completed, we utilize HDBSCAN[24] to perform clustering. HDBSCAN is an advanced clustering algorithm capable of handling variable densities. Since TAD regions do not have a fixed cluster size and exhibit varying densities with hierarchical properties, HDBSCAN is well-suited for this task compared to other clustering algorithms. We input the embedded data from the previous step into the HDBSCAN algorithm to identify potential TAD regions. The output from HDBSCAN consists of clusters of varying sizes, and not all clusters are considered TADs. To determine TAD size, we focus on regions ranging from 100 Kb to 5 Mb, which corresponds to the typical TAD size in mammalian organisms[8]. Clusters within this size range are labeled as TADs, while clusters outside this range are discarded.

### NetMF

We utilized NetMF[27] in our pipeline; a matrix factorization embedding framework in closed form and demonstrated improvement compared to other embedding framework such as DeepWalk[42], LINE[43], node2vec[44], etc. We directly incorporated the NetMF implementation from Rozemberczki et al.[45] into EmbedTAD and we implemented it's GPU compatible version with pytorch and cugraph. Qiu et al.[27] explained LINE, PTE[46], DeepWalk and node2vec's objective function as factorized matrix in closed form, and proposed a network embedding, NetMF, based on the DeepWalk matrix, as it is both computationally efficient and a more generalized formulation. Consider a graph, $G = (V, E, A)$ and its properties, $A \in \mathbb{R}_+^{|V| \times |V|}$ = adjacency matrix, diagonal matrices, $\Delta_{row} = diag(Ae)$, $\Delta_{col} = diag(A^{W_s}e)$ and $\Delta = diag(a_1, \dots, a_{|V|})$, undirected graph where $\Delta_{row} = \Delta_{cob}$ $vol(G) = \sum_i \sum_j A_{ij} = \sum_i a_i$, $a_i$ = generalized degree of vertex $i$, $W_s$ = window size, and $n_s$ = # of negative sampling. Based on this properties, LINE expressed their objective function as maximization problem,

$$\mathbb{L} = \sum_i^{|V|} \sum_j^{|V|} A_{ij} \left( \log \sigma \left( x_i^T y_j \right) + n_s E_{j' \sim N_d} \left[ \log \sigma \left( -x_i^T y_{j'} \right) \right] \right) \tag{3}$$

where $X, Y \in \mathbb{R}^{|V| \times a}$ that is $x_i, y_i$ = rows and $i = 1, \dots, |V|$, $\sigma$ = sigmoid function, noise distribution, $N_d(j) \propto a_j^{\frac{3}{4}}$ and Qiu et al.[27] expressed this objective function, $\mathbb{L}$ (Equation (3)) as matrix factorization form,

$$\log \left( vol(G) \Delta^{-1} A \Delta^{-1} \right) - \log n_s = XY^T \tag{4}$$

PTE is another form of LINE algorithm assuming graph as multiple network Lapacian and factorized it consider graph, $G = (V_1 \cup V_2, E, A)$ as heterogeneous network, specifically bipartite network. They described objective function as maximization problem similer to LINE where $E \subseteq V_1 \times V_2$, $A \in \mathbb{R}_+^{|V_1| \times |V_2|}$, and $vol(G) = \sum_i^{|V_1|} \sum_j^{|V_2|} A_{ij}$. Similar to LINE (Equation (3)), PTE's objective function is also expressed as matrix factorization in closed form,

$$\log\left(vol(G)\Delta_{row}^{-1} A \Delta_{col}^{-1}\right) - \log n_s = XY^T \tag{5}$$

DeepWalk is an implicit matrix factorization and primarily based on skip-gram negative sampling (SGNS). Following Levy et al.[47], SGNS implicitly expressed as

$$\log\left(\frac{\#(w_c, c_w)|\mathcal{C}|}{\#(w_c).\#(c_w)}\right) - \log n_s \tag{6}$$

where $C$ = corpus, $w_c$ = corpus of words and $c_w$ = context. Qiu et al.[27] showed that DeepWalk produced low-rank transformation of normalized Lapacian matrix in closed form and LINE (second order proximity) is a special case of DeepWalk implementation where window size, $W_s = 1$. For infinite random walk, $R_w \rightarrow \infty$, Equation (6) expresses DeepWalk objective function in closed form matrix factorization

$$\log\left(\frac{vol(G)}{W_s}\left(\sum_r^{W_s} \rho^r\right)\Delta^{-1}\right) - \log n_s \tag{7}$$

where $\rho = \Delta^{-1}A$. Considering transition probability tensor, $\underline{\tau}^r$, $\mu = w_{c_{i-1}}^n$ and stationary distribution, $\pi$, Qiu et al.[27] expressed node2vec in closed form

$$\frac{\#(w_c, c_w)|\mathcal{C}|}{\#(w_c).\#(c_w)} \xrightarrow{param} \frac{1}{2W_s}\sum_r^{W_s}\frac{\left(\sum_\mu \pi_{w_c\mu}\underline{\tau}_{c_w w_c\mu}^r + \sum_\mu \pi_{c_w\mu}\underline{\tau}_{w_c c_w\mu}^r\right)}{\left(\sum_\mu \pi_{w_c\mu}\right)\left(\sum_\mu \pi_{c_w\mu}\right)} \tag{8}$$

Based on the closed-form matrix factorization analyses of other embedding algorithms (Equations (4), (5), (7), (8)), Qiu et al.[27] developed a network embedding algorithm, NetMF, which unified these embedding approaches under a single baseline. Among the matrices considered, the DeepWalk matrix is more general and computationally efficient, and NetMF was designed with careful consideration of the skip-gram theoretical foundation, negative sampling, and the DeepWalk matrix. Qiu et al.[27] proposed NetMF for both small and large context window sizes, where the key difference lies in matrix definition. For small window sizes, they directly used the DeepWalk matrix,

$$\chi = \frac{vol(G)}{n_s W_s}\left(\sum_r^{W_s} \rho^r\right)\Delta^{-1} \tag{9}$$

while for large window sizes, they approximated a matrix for calculation efficiency. For small window size, $W_s$, NetMF computes DeepWalk matrix, $\chi$ and then defines,

$$\chi' = max(\chi, 1) \tag{10}$$

inspired by Shifted PPMI approach[47]. Since the direct computation of $\log \chi$ is difficult and expensive, they proposed $\chi'$, calculated $\log \chi'$ and applied Singular Value Decomposition (SVD) to factorize :

$$\log \chi' = U_a \Sigma_a V_a^T \tag{11}$$

Finally, the Rank-a network embedding is generated as: $U_a\sqrt{\sum_a}$. Qiu et al.[27] also showed that matrix, $\chi$ has a close relationship with normalized graph Laplacian which simplifies the computation for large window, $W_s$.

Specifically, they approximated the top-$h$ eigenpairs as

$$\Delta^{-\frac{1}{2}}A\Delta^{-\frac{1}{2}} \approx U_h\Lambda_h U_h^T \tag{12}$$

Then, they computed the approximate matrix as:

$$\hat{\chi} = \frac{vol(G)}{n_s}\Delta^{-\frac{1}{2}}U_h\left(\frac{1}{W_s}\sum_r^{W_s}\Lambda_h^r\right)U_h^T\Delta^{-\frac{1}{2}} \tag{13}$$

The remaining steps follow the same procedure as in the small window case to generate embeddings of the input graph using Equation (10) for large window size. Detailed proofs of each equation can be found in[27].

## HDBSCAN

HDBSCAN[24] is a hierarchical density-based clustering method that addresses several issues with well-known density and hierarchical-based clustering algorithms such as DBSCAN[41], DENCLUE[48], and OPTICS[49]. Based on the global density threshold, the majority of density-based algorithms, including DBSCAN, cluster data non-hierarchically. For their globally defined density threshold, they are unable to accurately characterize nested densities or variable densities[24]. Additionally, other density or hierarchically based clustering algorithm needs sensitive parameters such as DBSCAN requires two inputs: *Eps* = maximum distance between two points, and *minPts* = minimal number of points in a neighborhood. It can be challenging to determine the ideal value for these parameters, particularly when dealing with big datasets like genomic data. These limitations are addressed by HDBSCAN, which also offers greater flexibility in identifying clusters across a range of datasets[24]. Moreover, HDBSCAN yields more stable clusters from a tree of discovered clusters which is another advantage over other hierarchical based clustering algorithms. They approached this challenge as a maximization problem and optimized it by taking into account the optimal smallest number of cluster points that might be found. HDBSCAN[24] is an extended version of DBSCAN[41] and OPTICS[49] algorithm. HDBSCAN identifies clusters from different densities adjusting the epsilon value. It generates a tree data structure to find the significant hierarchical cluster and introduces long-term stability mechanism of clusters. Initially, HDBSCAN computes a set of core distances considering all points expressed as $\mu$, $\Delta_c(x_p)|\Delta_c(x_p) \leq \varepsilon$ where $x_p \in X$, $\mu$ = minimum number of points, $\varepsilon$ = maximum distance between two points, and calculates mutual reachability distance from the derived set of core distances:

$$\nabla(x_p, x_q) = max\left\{\Delta_c(x_p),\ \Delta_c(x_q),\ \Delta_c(x_p, x_q)\right\} \tag{14}$$

Next, it builds a Minimum Spanning Tree (MST) using the mutual reachability distance where it makes a disjoint small tree and iteratively adds lower weighted edges. It takes the weight of the corresponding object and adds self-loops in each node. After creating MST, it creates a dendrogram to create the clusters. It starts from the root as a single cluster and iteratively assigns other clusters to its descendant nodes which are sorted by weight in descending order.

## Output

EmbedTAD's Output module integrates all TAD regions generated by the Clustering module. During the preprocessing step, we divided the Hi-C contact matrix into sub-matrices to accelerate graph generation and processing. For each sub-matrix, EmbedTAD's Clustering module identifies TAD regions corresponding to a specific portion of the Hi-C contact matrix. However, to obtain continuous and distinct TADs across the entire matrix, we remove overlapping or redundant TADs in the Q regions (Fig. 1B).

As described earlier, in the preprocessing step, we extended each sub-matrix $p_{i+1} \times p_{i+1}$ by $q = 3$ Mb from the previous sub-matrix $p_i \times p_i$. This extension can result in overlapping or duplicate TADs in the Q region shared between the two sub-matrices. To resolve this, we calculate the

TAD Quality (TQ) score[12] for each overlapping $Q$ region in both $p_i \times p_i$ and $p_{i+1} \times p_{i+1}$. The TQ score works by maximizing intra-TAD interactions while minimizing inter-cluster interactions, ensuring that TADs are well-defined. We then retain the TADs from the sub-matrix with the higher TQ score. This selection process is defined by the following equations:

$$Q^{TAD} = (p_i \times p_i)^Q \cup (p_{i+1} \times p_{i+1})^Q \tag{15}$$

$$Q^{TAD} \in \{(p_i \times p_i)^Q\}^{TAD} = p_i^{TAD} \tag{16}$$

$$Q^{TAD} \in \{(p_{i+1} \times p_{i+1})^Q\}^{TAD} = p_{i+1}^{TAD} \tag{17}$$

$$f(p_i^{TAD}) = TQ_i \tag{18}$$

$$f(p_{i+1}^{TAD}) = TQ_{i+1} \tag{19}$$

$$Q_f^{TAD} = \begin{cases} p_i^{TAD}, & \text{if } TQ_i > TQ_{i+1} \\ p_{i+1}^{TAD}, & \text{otherwise} \end{cases} \tag{20}$$

Equation (15) defines the union of TADs from both overlapping sub-matrices. Equations (16) and (17) assign TADs from each respective sub-matrix. Equations (18) and (19) calculate the TAD Quality scores for each sub-matrix, and finally, Equation (20) selects the TADs from the sub-matrix with the higher TQ score to ensure the best representation in the overlapping $Q$ region.

After resolving overlapping or redundant TADs, we merge the retained TADs from both the previous and current sub-matrices to produce continuous, distinct TAD regions. This ensures comprehensive coverage of the entire contact matrix, including boundary regions, while maintaining high TAD Quality scores.

The final output of EmbedTAD is provided in a BED-like format, representing the start and end positions of each detected TAD region.

## Statistics and Reproducibility
Result and method section includes our experiment's statistics and reproducibility. EmbedTAD was assessed using in-silico Hi-C dataset, and our pipeline was once more verified using real Hi-C dataset. We used various noise levels of in-silico Hi-C data to assess EmbedTAD with SI, DBI, and CHI. We evaluated EmbedTAD's performance using MoC and TAD Quality score. From the in-silico Hi-C dataset, we contrasted True and EmbedTAD-detected TAD. Using the real Hi-C dataset, we examined the $TADadjR^2$ score and average ChIP-seq signal across several organisms and resolutions. From the real Hi-C dataset, we assessed the quantity of TADs, the distribution of TAD sizes, and the IS of EmbedTAD-detected TADs at various resolutions and organisms. We used a variety of resolutions and organisms with distinct ChIP-seq signals including CTCF, RAD21, H3K27ac to biologically validate EmbedTAD's detected TADs from genuine Hi-C datasets. The data, tables, and figures in this manuscript and supplemental information file contain all of our analyses.

## Reporting summary
Further information on research design is available in the Nature Portfolio Reporting Summary linked to this article.

## Data availability
In-silico Hi-C was downloaded from HiCToolsCompare. Hi-C contact maps of human lymphoblastoid cell (GM12878) and mouse lymphoma cell (CH12LX) were downloaded from NCBI GEO GSE63525[3]. ChIP-seq signal data were downloaded from https://www.encodeproject.org/, https://hgdownload.soe.ucsc.edu/goldenPath/hg19/database/and https://hgdownload.soe.ucsc.edu/goldenPath/mm9/database/. mESC data downloaded from GSE35156[8]. Mus musculus data were downloaded from

GSE210418[40]. We used https://hicexplorer.readthedocs.io/en/latest/index.html, https://github.com/kmiles18/TAD-callers-comparison, https://github.com/vaquerizaslab/tadtool, and https://github.com/XiaoTaoWang/TADLib?tab=readme-ov-fileto plot some of our analysis results. The source data for experimental results and analysis data is found at https://github.com/OluwadareLab/EmbedTAD/tree/main/ra_data.

## Code availability
The EmbedTAD source code is freely available at https://github.com/OluwadareLab/EmbedTAD. The EmbedTAD documentation is available at: https://github.com/OluwadareLab/EmbedTAD/wiki.

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

## Acknowledgements

This work was supported by the National Institutes of General Medical Sciences of the National Institutes of Health under award number R35GM150402 to O.O.

## Author contributions

H.M.A.M.C. conducted the analysis, wrote, and revised the manuscript and O.O. conceived, wrote, revised the manuscript, and supervised this project. All authors reviewed the manuscript.

## Competing interests

The authors declare no competing interests.
