## [Transparent Peer Review file · Communications Biology]

EmbedTAD: Using Graph Embedding and Unsupervised Learning to Identify TADs from High-Resolution Hi-C Data

Corresponding Author: Dr Oluwatosin Oluwadare

Version 0:

Reviewer comments:

Reviewer #1

(Remarks to the Author)

The manuscript proposes EmbedTAD, TAD caller over Hi-C dataset. Even though there are now almost more than 30 TAD callers, the work suggests a graph and clustering-based TAD caller which includes intellectual merits.

Below are my comments:

Major:

- 1- Introduction section is relatively short. Related work part should be expanded. Instead of citing almost 30 TAD callers, authors may cite the following TAD comparison papers:
 - A comparison of topologically associating domain callers over mammals at high resolution E Sefer BMC bioinformatics 23 (1), 127
 - Forcato M, Nicoletti C, Pal K, Livi CM, Ferrari F, Bicciato S. Comparison of computational methods for hi-c data analysis. Nat Methods. 2017;14(7):679–85.
- 2- What is the advantage of using HDBSCAN instead of DBSCAN?
- 3- Can you describe metrics such as TADadjR2 explicitly?
- 4- How does NetMF unify the given set of GNN-based approaches as said in the following sentence? 'We utilized NetMF [21], a network matrix factorization method for network embedding, unifying LINE [36], DeepWalk [37], PTE [38], and node2vec [39], while 380 considering negative sampling.'
- 5- How does the method work on PLAC-seq data which is not interaction data?
- 6- In Figure 5A, why are chromosomes 15 and 18 empty?
- 7- What are EmbedTAD's advantages over the existing TAD callers?
- 8- Can you mathematically describe NetMF? Especially how MAX is obtained, etc.

Minor:

- 1- Can you increase the resolution of Figure 1? Can you separate A and B from rest of them since A and B are related to the method, but the rest of them are results?
- 2- In addition to biological validation, can we also validate the performance on semi-synthetic data by traditional classification performance metrics?

Reviewer #2

(Remarks to the Author)

In this work, the authors present EmbedTAD, a novel method to determine TAD locations and boundaries using a graph embedding technique. Given Hi-C data, the method can determine TAD locations and boundaries well. They compare their method with many of the leading methods in the literature, like TopDom, and the results are impressive. The authors validate the boundary locations using CTCF and histone modification data. Going beyond Hi-C data, the authors use EmbedTAD to recover TADs from H3K4me3 and PLAC-seq data. This enables them to predict how TAD rearrangement happens during cell differentiation.

Overall, this is a very good study, and I recommend publishing it in Communications Biology. Authors may consider addressing the following comments/questions before publication:

1) The figure captions can be greatly improved. In many figures, the caption does not explain what is shown. For example, in Figure 3G and H what are the bottom figures?

2) In many locations, EmbedTAD shows impressive agreement compared to existing methods. However, there are some regions where it seems to deviate from other leading methods. The authors are recommended to discuss those regions in more detail.

3) When authors say "High-resolution" data, is it often 10kb resolution. However, CTCF boundary locations are typically smaller than this. How do authors reconcile this?

Version 1:

Reviewer comments:

Reviewer #1

(Remarks to the Author)

The authors have addressed all my comments.

Response to Reviewer Comments

Reviewer #1 (Remarks to the Author):

The manuscript proposes EmbedTAD, TAD caller over Hi-C dataset. Even though there are now almost more than 30 TAD callers, the work suggests a graph and clustering-based TAD caller which includes intellectual merits.

Below are my comments:

Major:

1- Introduction section is relatively short. Related work part should be expanded. Instead of citing almost 30 TAD callers, authors may cite the following TAD comparison papers:

- A comparison of topologically associating domain callers over mammals at high resolution
E Sefer BMC bioinformatics 23 (1), 127

- Forcato M, Nicoletti C, Pal K, Livi CM, Ferrari F, Bicciato S. Comparison of computational methods for hi-c data analysis. Nat Methods. 2017;14(7):679–85.

Thank you for your thoughtful comment. In response, we have completely revised the Introduction section, including the related work paragraph, and included necessary citations to acknowledge previous studies. We highlighted all the changes in red text in the introduction section of this revised manuscript.

2- What is the advantage of using HDBSCAN instead of DBSCAN?

We appreciate your comment. To answer this question satisfactorily, in this revised manuscript, we have revised Section 4.2.2 of the HDBSCAN to include its advantages compared to DBSCAN algorithms and other *hierarchical-based clustering algorithms*. All changes are highlighted in red text in this revised manuscript.

3- Can you describe metrics such as TADadjR2 explicitly?

Thank you for your comment. In this revised manuscript, we have provided a detailed description of this metric in Section 2.5 to explicitly explain how it is being calculated and the interpretation of values. All changes are highlighted in red text in this revised manuscript.

4- How does NetMF unify the given set of GNN-based approaches as said in the following sentence? 'We utilized NetMF [21], a network matrix factorization method for network embedding, unifying LINE [36], DeepWalk [37], PTE [38], and node2vec [39], while considering negative sampling.'

Thank you for your valuable comment. In this revised manuscript, we have addressed the question of how NetMF unifies LINE, DeepWalk, PTE, and node2vec while considering negative sampling. We have provided a detailed step-by-step explanation of the rationale and how this unification is achieved in Section 4.2.1.

In their study, Qiu et al., 2018, explained the objective functions of LINE, PTE, DeepWalk, and node2vec as closed-form matrix factorizations and proposed a new network embedding

method, NetMF, based on the DeepWalk matrix, as it is both computationally efficient and a more generalized formulation.

In our explanation, we first broke down each of these algorithms and presented their corresponding equations. Finally, we described how NetMF unifies the principles of these algorithms, beginning in Line 507. All changes in this revised manuscript are highlighted in red text.

1. Qiu, J., Dong, Y., Ma, H., Li, J., Wang, K., & Tang, J. (2018, February). Network embedding as matrix factorization: Unifying deepwalk, line, pte, and node2vec. In *Proceedings of the eleventh ACM international conference on web search and data mining* (pp. 459-467).

5- How does the method work on PLAC-seq data which is not interaction data?

Thank you for your comment. We would like to clarify that, as shown by Fang et al., 2016, Proximity Ligation-Assisted ChIP-seq (PLAC-seq) is not limited to ChIP-seq signals but is capable of capturing long-range chromatin interactions. In this revised manuscript, we have provided details on the PLAC-seq data's ability to capture long-range interactions in Section 2.9, and its uses to capture TADs in other studies [Rosen et al., 2023; Lee et al., 2024]. All changes are highlighted in red text in this revised manuscript.

Excerpt from Section 2.9:

In their study, Fang et al., 2016 demonstrated that PLAC-seq improves both efficiency and accuracy over ChIA-PET in detecting long-range chromatin interactions in mammalian cells, generating reproducible contact maps across biological replicates. In mouse embryonic stem (ES) cells, PLAC-seq successfully captured promoter-centered interactions, and H3K4me3 PLAC-seq proved useful for identifying chromatin interactions at active or poised promoters [Fang et al., 2016].

1. Fang, R., Yu, M., Li, G., Chee, S., Liu, T., Schmitt, A. D., & Ren, B. (2016). Mapping of long-range chromatin interactions by proximity ligation-assisted ChIP-seq. *Cell research*, 26(12), 1345-1348.
2. Lee, D., Kang, J., & Kim, A. (2024). TAD-dependent sub-TAD is required for enhancer–promoter interaction enabling the β -globin transcription. *The FASEB Journal*, 38(22), e70181.
3. Rosen, J., Lee, L., Abnousi, A., Chen, J., Wen, J., Hu, M., & Li, Y. (2023). HPTAD: A computational method to identify topologically associating domains from HiChIP and PLAC-seq datasets. *Computational and Structural Biotechnology Journal*, 21, 931-939.

6- In Figure 5A, why are chromosomes 15 and 18 empty?

Thank you for your great comment and observation. In this revised manuscript, we have provided a detailed description of this question in Section 2.9. All changes are highlighted in

red text in this revised manuscript. We have also provided excerpt from the Section 2.9 for the reason for this below following up on the question 5 above.

Excerpt from Section 2.9:

In their study, Fang et al., 2016 demonstrated that PLAC-seq improves both efficiency and accuracy over ChIA-PET in detecting long-range chromatin interactions in mammalian cells, generating reproducible contact maps across biological replicates. In mouse embryonic stem (ES) cells, PLAC-seq successfully captured promoter-centered interactions, and H3K4me3 PLAC-seq proved useful for identifying chromatin interactions at active or poised promoters [Fang et al., 2016].

Since Rosen et al., 2023 developed cutting-edge techniques for identifying TADs using mESC H3K4me3 PLAC-seq data, we utilized their TADs as the ground truth to evaluate our method. We calculated EmbedTAD's recovery rate using its detected TADs from mESC bulk Hi-C data. As shown in Figure 6A, EmbedTAD recovered approximately 67% of the TAD regions identified in the PLAC-seq data, with the exception of chromosomes 15 and 18 at 40 Kb resolution. While our method successfully recovered TADs on most chromosomes, it struggled on chromosomes 15 and 18 where PLAC-seq coverage was particularly sparse, leading to weaker domain signals. Because PLAC-seq targets specific proteins such as H3K4me3, the resulting data are inherently sparser than Hi-C and better suited for detecting chromatin loops (e.g., promoter–enhancer and promoter–promoter interactions) rather than large-scale 3D domains [Fang et al., 2016]. This reflects a limitation of the input data rather than the algorithm itself, as bulk Hi-C provides the broader coverage needed for robust TAD detection and chromosome structure analysis.

1. Fang, R., Yu, M., Li, G., Chee, S., Liu, T., Schmitt, A. D., & Ren, B. (2016). Mapping of long-range chromatin interactions by proximity ligation-assisted ChIP-seq. *Cell research*, 26(12), 1345-1348.
2. Rosen, J., Lee, L., Abnousi, A., Chen, J., Wen, J., Hu, M., & Li, Y. (2023). HPTAD: A computational method to identify topologically associating domains from HiChIP and PLAC-seq datasets. *Computational and Structural Biotechnology Journal*, 21, 931-939.

7- What are EmbedTAD's advantages over the existing TAD callers?

Thank you for your valuable comment. We have described the advantages of EmbedTAD in the Introduction and Discussion sections and demonstrated strong analysis results throughout the manuscript to highlight our method's strengths. To clarify the importance of EmbedTAD, we have revised the Introduction and Discussion sections, adding explicit statements in red text for the reader's convenience. All changes are highlighted in red text in this revised manuscript.

For easy readability and to answer this question directly, here we clarify the advantages of EmbedTAD over other TAD callers in detail below:

1. **Novel approach** – EmbedTAD proposes a new method for TAD detection by representing chromatin interactions as graphs. Section 2.1, 4, and Figure 1

2. **Network embedding** – Uses network embedding to reduce graph data to a lower-dimensional space, the first to our knowledge in bulk Hi-C analysis. Section 2.1, 4, 4.2.1, and Figure 1
3. **Competitive performance** – Demonstrates strong performance compared to state-of-the-art (SOTA) TAD callers using both in-silico and real Hi-C data. Section 2.2, 2.3, 2.4, 2.5, Figure 2G, 3, Supplementary Figure S3, S4, S5, S6, S7, S8, S9, S18, and Supplementary Table S2, S5
4. **Robustness to noise** – Shows consistent results across different noise levels in in-silico Hi-C data with stable hyperparameters. Section 2.2, 2.3, Figure 2, Supplementary Figure S1, S2, S3, S18, and Supplementary Table S5
5. **Dual implementation** – Provides both CPU and GPU versions, enabling broad usability while maintaining low memory consumption, reduced runtime, and consistency across implementations. Section 2.7, 4.2.1, Supplementary Figure S1 and Supplementary Table S3, S4.
6. **Statistical validation** – Outperforms other SOTA TAD callers in multiple statistical analyses on real Hi-C data. Section 2.4, 2.5, 2.6, Figure 3, Supplementary Figure S4, S5, S6, S7, S8, S9, and Supplementary Table S2
7. **Biological validation** – Demonstrates strong biological consistency, including protein site verification and PLAC-seq recovery. Section 2.8, 2.9, Figure 5, 6A, 6B, 6C, 6D, 6E, and Supplementary Figure S12, S13, S14, S15, S16, S17
8. **Real-world application** – Successfully applied to T-cell development, showcasing its practical biological relevance. Section 2.10, and Figures 6F, 6G, 6H, 6I.

Overall, EmbedTAD uniquely combines novelty, efficiency, robustness, and superior validation, establishing it as a strong TAD caller over existing methods.

8- Can you mathematically describe NetMF? Especially how MAX is obtained, etc.

Thank you for your valuable comment. In this revised manuscript, we have rewritten the entire section on NetMF to address some of the reviewers' questions about how NetMF unifies LINE and Deepwalk, among other things. Therefore, we have provided a clearer explanation of how the MAX value is obtained and used in the NetMF algorithm in Section 4.2.1 (Equations 9 and 10). All changes are highlighted in red in this revised manuscript.

Minor:

1- Can you increase the resolution of Figure 1? Can you separate A and B from rest of them since A and B are related to the method, but the rest of them are results?

Thank you for your great comment. In this revised manuscript, we have separated Figure 1 into two distinct figures; Figure 1 explains the EmbedTAD pipeline, and Figure 2 explains the analysis results, which were in Figure 1. We also increased the resolution of these two figures in accordance with the journal's standard suggestion (<https://www.nature.com/commsbio/submit/submission-guidelines>). We have also presented Figure 1 as a Landscape image for easier viewing. We highlighted the captions of the segregated figures in red text in the main manuscript.

2- In addition to biological validation, can we also validate the performance on semi-synthetic data by traditional classification performance metrics?

Thank you for your comment. In this revised manuscript, we have added two metrics, Adjusted Mutual Information (AMI) and Adjusted Rand Index (ARI), for performance evaluation on synthetic data, based on their usage in previous TAD detection methods [Li et al., 2021 and Liu et al., 2024]. Our observations are described in Sections 2.3, as well as in the Supplementary Materials (Subsection 1.1.4). All changes in the revised manuscript are shown in red text. For convenience, the content of Supplementary Subsection 1.1.4 describing our observations is provided below:

Since TAD borders are enriched with numerous biological landmarks, including CTCF and histone markers, TAD detection techniques are typically assessed using biological studies to validate their effectiveness [Dixon et al., 2012]. The lack of ground truth for actual datasets to assess a TAD caller's performance is another factor. We used Forcato et al.'s, 2017 synthetic dataset, which enabled us to assess the effectiveness of our approach using classical analysis. We evaluated our approach using Adjusted Mutual Information (AMI) and Adjusted Random Index (ARI) analysis, based on their usage in previous TAD detection methods [Li et al., 2021 and Liu et al., 2024]. Predicted bins may vary but generally overlap with the actual ground truth in terms of synthetic data, and TAD regions identified by TAD callers are neither entirely similar nor exact in terms of bins. We select AMI and ARI to assess clustering quality in light of this pattern. AMI, which has a value between 0 (random match) and 1 (perfect match), calculates the cluster similarity while taking into account the random assignment of clusters. ARI, a different metric that modifies assignment randomness and ranges from -1 (worst) to 1 (perfect), compares the bins to determine whether or not they are in the same cluster. Since there may be numerous biological landmarks, these metrics measure the TAD boundary and domain assignment accuracy, which are essential for TAD identification problem and require additional biological validation. EmbedTAD showed competitive performance over other state-of-the-art TAD callers in terms of AMI and ARI scores (Figure S18, Table S5). We found that EmbedTAD showed nearly the same mean and median values across various noise levels in this conventional assessment metric, demonstrating that our approach generates highly symmetrical values for the same synthetic Hi-C data across various noise levels. Furthermore, we found no outliers in this metrics test, demonstrating reproducible, scalable, and consistent outcomes across a range of noise levels. These findings also demonstrate that our network embedding and graph data format regularly lower the noise sensitivity. In comparison with traditional state-of-the-art techniques, this observation indicates that our approach is competitive, robust and consistent across a range of noise levels and attained a novel position by introducing new technology in the TAD detection problem.

Figure S18: AMI and ARI scores using synthetic Hi-C data. Boxplot shows the mean and median values of AMI and ARI scores, where EmbedTAD showed competitive results without any outliers and nearly identical mean and median values.

AMI				
Algorithm	Min	Mean	Median	Max
Armatus	0.969998668	0.981896467	0.982218487	0.988787808
Caspian	0.747592396	0.868172879	0.883509127	0.932162521
ClusterTAD	0.772843868	0.812828406	0.811367587	0.854346679
EmbedTAD	0.900536621	0.910965023	0.910481888	0.921687682
HiCSeq	0.753674818	0.886086799	0.901944653	0.965147032
IC-Finder	0.764436932	0.833936021	0.84649006	0.86612884
Spectral	0.861384586	0.876066615	0.875416135	0.889114115
TopDom	0.959487214	0.970948198	0.970439816	0.987151912
ARI				
Algorithm	Min	Mean	Median	Max
Armatus	0.920425007	0.959321786	0.959641924	0.98590165
Caspian	0.180508127	0.556910953	0.60205588	0.801420875
ClusterTAD	0.241049787	0.38545023	0.38899107	0.496041101
EmbedTAD	0.656675832	0.706174672	0.704519724	0.757862566
HiCSeq	0.39822954	0.673875186	0.696360851	0.882196289

IC-Finder	0.211452146	0.504200175	0.56672881	0.628566333
Spectral	0.582321521	0.617654169	0.617416685	0.652944626
TopDom	0.893884355	0.922603355	0.922048938	0.965793155

Table S5: AMI and ARI scores using synthetic Hi-C data. Min, mean, median, and max values of AMI and ARI scores, where EmbedTAD showed competitive results without any outliers and nearly identical mean and median values.

1. Forcato, M., Nicoletti, C., Pal, K., Livi, C. M., Ferrari, F., & Bicciato, S. (2017). Comparison of computational methods for Hi-C data analysis. *Nature methods*, 14(7), 679-685.
2. Li, X., Zeng, G., Li, A., & Zhang, Z. (2021). DeTOKI identifies and characterizes the dynamics of chromatin TAD-like domains in a single cell. *Genome biology*, 22(1), 217.
3. Liu, E., Lyu, H., Liu, Y., Fu, L., Cheng, X., & Yin, X. (2024). Identifying TAD-like domains on single-cell Hi-C data by graph embedding and changepoint detection. *Bioinformatics*, 40(3), btae138.
4. Dixon, J. R., Selvaraj, S., Yue, F., Kim, A., Li, Y., Shen, Y., ... & Ren, B. (2012). Topological domains in mammalian genomes identified by analysis of chromatin interactions. *Nature*, 485(7398), 376-380.

Reviewer #2 (Remarks to the Author):

In this work, the authors present EmbedTAD, a novel method to determine TAD locations and boundaries using a graph embedding technique. Given Hi-C data, the method can determine TAD locations and boundaries well. They compare their method with many of the leading methods in the literature, like TopDom, and the results are impressive. The authors validate the boundary locations using CTCF and histone modification data. Going beyond Hi-C data, the authors use EmbedTAD to recover TADs from H3K4me3 and PLAC-seq data. This enables them to predict how TAD rearrangement happens during cell differentiation.

Overall, this is a very good study, and I recommend publishing it in Communications Biology. Authors may consider addressing the following comments/questions before publication:

- 1) The figure captions can be greatly improved. In many figures, the caption does not explain what is shown. For example, in Figure 3G and H what are the bottom figures?

Thank you for your comment. In this revised manuscript, we have revised all our figure labels to make them more descriptive and easier to understand, in addition to revising Figures 4G and 4H (previously Figures 3G and 3H). It is worth noting that we have split the previous Figure 1 into Figures 1 and 2 in this revised manuscript for improved visualization. All the changes in this revised manuscript are highlighted in red.

2) In many locations, EmbedTAD shows impressive agreement compared to existing methods. However, there are some regions where it seems to deviate from other leading methods. The authors are recommended to discuss those regions in more detail.

We thank the reviewer for this valuable comment. We made the same observation in Figures 3A–C and investigated the underlying cause. Upon careful review, we realized that our original boundary selection code was only counting one boundary per TAD, particularly when two different TADs started and ended at the same bin, rather than considering the left and right boundaries independently [Dixon et al., 2012; Gong et al., 2018]. After correcting this oversight, our updated plots show consistency between EmbedTAD and other leading methods, with improved alignment of boundary positions.

EmbedTAD consistently showed enrichment around boundaries, mostly to other TAD callers across analyses. We also note that methods detecting a larger number of TADs (Supplementary Figure S19) naturally have a higher likelihood of capturing additional boundary variations, which partly explains why the other methods showed slightly higher enrichment of CTCF, RAD21, and SMC3. Taken together, these results underscore the robustness of our algorithm and provide a clear explanation of ChIP-seq signal enrichment around TAD boundaries, which is essential for accurately identifying TADs and maintaining biological information for future analysis. These results underscore the robustness of our algorithm and provide a clear context for the deviations noted in the initial plots. We sincerely appreciate the reviewer for bringing this to our attention.

We have revised our manuscript to explain our results in Section 2.4 and included the updated Figures, in Figure 3A-C, and the corresponding Supplementary Figures S4-S6. We have added GM12878 chromosome 19 at 10Kb resolution CTCF enrichment and TAD count plot here as a reference.

- Dixon, J. R., Selvaraj, S., Yue, F., Kim, A., Li, Y., Shen, Y., ... & Ren, B. (2012). Topological domains in mammalian genomes identified by analysis of chromatin interactions. *Nature*, 485(7398), 376-380.
- Gong, Y., Lazaris, C., Sakellaropoulos, T., Lozano, A., Kambadur, P., Ntziachristos, P., ... & Tsiganos, A. (2018). Stratification of TAD boundaries reveals preferential

insulation of super-enhancers by strong boundaries. *Nature communications*, 9(1), 542.

3) When authors say "High-resolution" data, is it often 10kb resolution. However, CTCF boundary locations are typically smaller than this. How do authors reconcile this?

We thank the reviewer for raising this important point. In our work, we used Hi-C matrices at 5 Kb and 10 Kb resolution. Following conventions in the literature, we define any resolution ≥ 10 Kb as "high-resolution" in the context of Hi-C analysis [Fang et al., 2024; Hua et al., 2024; Hong et al., 2020].

We fully agree with the reviewer that CTCF binding sites are much smaller, typically a few hundred base pairs [Kim et al., 2007; Phillips et al., 2009; Ong et al., 2014]. However, a single CTCF site does not define a TAD boundary on its own. Rather, TAD boundaries are broad regions, ranging from hundreds of Kb up to several Mb, and are enriched in clusters of proteins such as CTCF and cohesin [Dixon et al., 2012; Rao et al., 2014].

To reconcile these scales, we follow the established approach in the TAD literature [An et al., 2019; Hua et al., 2024; Xu et al., 2024; Zufferey et al., 2018] by binning Hi-C matrices (e.g., into 5 Kb or 10 Kb bins) and mapping CTCF or other ChIP-seq signals to their corresponding bins. For example, if a Hi-C interval is divided into 10 Kb bins, CTCF binding events at base-pair resolution are aggregated into the appropriate bin, which allows their enrichment to be compared against TAD boundaries detected at the bin level, e.g., if CTCF is at position 12,350 bp, and we are using 10 Kb bins, it falls into bin 2 = 10,001–20,000 bp. This allows you to measure whether protein enrichment occurs at predicted TAD boundaries, even though the original signals are at a higher resolution.

Using this strategy, we validated our predicted TAD boundaries against multiple protein signals (CTCF, SMC3, RAD21, etc.) and consistently observed enrichment at the detected boundary regions. This supports the robustness of our approach, even though the protein binding sites themselves occur at base-pair resolution.

1. Kim, T. H., Abdullaev, Z. K., Smith, A. D., Ching, K. A., Loukinov, D. I., Green, R. D., ... & Ren, B. (2007). Analysis of the vertebrate insulator protein CTCF-binding sites in the human genome. *Cell*, 128(6), 1231-1245.
2. Phillips, J. E., & Corces, V. G. (2009). CTCF: master weaver of the genome. *Cell*, 137(7), 1194-1211.
3. Ong, C. T., & Corces, V. G. (2014). CTCF: an architectural protein bridging genome topology and function. *Nature Reviews Genetics*, 15(4), 234-246.
4. Dixon, J. R., Selvaraj, S., Yue, F., Kim, A., Li, Y., Shen, Y., ... & Ren, B. (2012). Topological domains in mammalian genomes identified by analysis of chromatin interactions. *Nature*, 485(7398), 376-380.
5. Rao, S. S., Huntley, M. H., Durand, N. C., Stamenova, E. K., Bochkov, I. D., Robinson, J. T., ... & Aiden, E. L. (2014). A 3D map of the human genome at kilobase resolution reveals principles of chromatin looping. *Cell*, 159(7), 1665-1680.

6. An, L., Yang, T., Yang, J., Nuebler, J., Xiang, G., Hardison, R. C., ... & Zhang, Y. (2019). OnTAD: hierarchical domain structure reveals the divergence of activity among TADs and boundaries. *Genome biology*, 20(1), 282.
7. Hua, D., Gu, M., Zhang, X., Du, Y., Xie, H., Qi, L., ... & Tian, D. (2024). DiffDomain enables identification of structurally reorganized topologically associating domains. *Nature Communications*, 15(1), 502.
8. Xu, J., Xu, X., Huang, D., Luo, Y., Lin, L., Bai, X., ... & Chen, H. (2024). A comprehensive benchmarking with interpretation and operational guidance for the hierarchy of topologically associating domains. *Nature Communications*, 15(1), 4376.
9. Zufferey, M., Tavernari, D., Oricchio, E., & Ciriello, G. (2018). Comparison of computational methods for the identification of topologically associating domains. *Genome biology*, 19(1), 217.
10. Fang, T., Liu, Y., Woicik, A., Lu, M., Jha, A., Wang, X., ... & Wang, S. (2024). Enhancing Hi-C contact matrices for loop detection with Capricorn: a multiview diffusion model. *Bioinformatics*, 40(Supplement_1), i471-i480.
11. Hong, H., Jiang, S., Li, H., Du, G., Sun, Y., Tao, H., ... & Bo, X. (2020). DeepHiC: A generative adversarial network for enhancing Hi-C data resolution. *PLoS computational biology*, 16(2), e1007287.